# CD44 Intracellular Domain: A Long Tale of a Short Tail

**DOI:** 10.3390/cancers15205041

**Published:** 2023-10-18

**Authors:** Spyros S. Skandalis

**Affiliations:** Biochemistry, Biochemical Analysis & Matrix Pathobiology Res. Group, Laboratory of Biochemistry, Department of Chemistry, University of Patras, 26504 Patras, Greece; skandalis@upatras.gr

**Keywords:** CD44, cancer, transcriptome, cell metabolism, cell growth, cell trafficking

## Abstract

**Simple Summary:**

This review is focused on the interactions of the CD44 cytoplasmic tail in order to unravel the high complexity of CD44 functions. CD44 serves as a cell surface receptor for various extracellular matrix molecules, mainly hyaluronan, and messenger molecules, such as growth factors, and has important functions in normal and disease states, the predominant one being cancer. CD44 coordinates both structural and signaling events through its highly conserved intracellular domain. Although short and devoid of any enzymatic activity, the CD44 intracellular domain possesses structural motifs that promote the interactions with cytoplasmic effectors involved in important cellular pathways, including cell trafficking, transcription, and metabolism, which regulate cellular functions like growth, survival, differentiation, stemness, and therapeutic resistance.

**Abstract:**

CD44 is a single-chain transmembrane receptor that exists in multiple forms due to alternative mRNA splicing and post-translational modifications. CD44 is the main cell surface receptor of hyaluronan as well as other extracellular matrix molecules, cytokines, and growth factors that play important roles in physiological processes (such as hematopoiesis and lymphocyte homing) and the progression of various diseases, the predominant one being cancer. Currently, CD44 is an established cancer stem cell marker in several tumors, implying a central functional role in tumor biology. The present review aims to highlight the contribution of the CD44 short cytoplasmic tail, which is devoid of any enzymatic activity, in the extraordinary functional diversity of the receptor. The interactions of CD44 with cytoskeletal proteins through specific structural motifs within its intracellular domain drives cytoskeleton rearrangements and affects the distribution of organelles and transport of molecules. Moreover, the CD44 intracellular domain specifically interacts with various cytoplasmic effectors regulating cell-trafficking machinery, signal transduction pathways, the transcriptome, and vital cell metabolic pathways. Understanding the cell type- and context-specificity of these interactions may unravel the high complexity of CD44 functions and lead to novel improved therapeutic interventions.

## 1. Introduction

The main function of cell surface transmembrane receptors is to sense extracellular signals and transduce them into an intracellular response. Most often, they bind to their specific ligand(s), leading to their conformational change and activation, which in turn alters their association with the cytoskeletal network and/or downstream signal transduction pathways to enable cells to respond to changes in their microenvironment. As such, CD44 is a single-chain transmembrane glycoprotein/part-time proteoglycan that belongs to the class of cell adhesion molecules (CAMs). The human gene encoding CD44 contains 19 exons and is located at chromosome 11p13. Exons 1 to 16 give rise to the extracellular domain (ectodomain) of CD44: the N-terminal signal sequence (exon 1), the link-homology hyaluronan-binding module (exons 2 and 3), and the stem region (exons 4–16). Exon 17 encodes the hydrophobic single-pass transmembrane domain, and exon 19 encodes the 73-amino-acid intracellular domain (ICD) or cytoplasmic tail of CD44 [1]. The inclusion of exon 18, normally absent in most CD44 transcripts, results in a short five-amino-acid cytoplasmic tail, which is generated by the use of an alternative translation stop codon [2]. Exons 1–5, 16, 17, and 19 are constant and common to all CD44 isoforms. The standard form of CD44 (CD44s) is encoded by these constant exons, while CD44 variant forms (CD44v) arise from the combination of constant exons with variable exons 7–15 (identified as v2–v10) into the stem region as a result of alternative mRNA splicing, giving CD44 proteins the potential for vast structural and functional diversity. Further post-translational modifications such as glycosylation with O-glycans, N-glycans, and glycosaminoglycans in specific CD44 isoforms add to the complexity and diversity of CD44 proteins. Importantly, the link-homology hyaluronan-binding domain, the transmembrane domain, and the cytoplasmic tail of CD44 show high similarity between species [3,4]. This remarkable sequence conservation indicates the important role of these domains in CD44 functions.

As a cell surface receptor, CD44 interacts with various extracellular matrix (ECM) molecules, such as hyaluronan (HA), osteopontin, collagen, fibronectin, and laminin, as well as growth factors and cytokines, MMPs, and serglycin proteoglycan [5]. Many of these interactions are cell-type-dependent and contribute to the plethora of CD44 functions. For example, CD44 can exist in “inactive” or “active” states according to the binding affinity to its main ligand HA [6]. In its “active” state, CD44 binds to HA, resulting in the promotion of cell migration, chemotaxis, rolling, and adhesion, as well as the organization of an HA-rich pericellular milieu and also the internalization and metabolism of HA. As a consequence, these CD44–HA interactions regulate several aspects of cellular behavior such as cell proliferation, growth, survival, differentiation, and pericellular matrix remodeling. However, the long list of functions of CD44 also include HA-independent activities. Importantly, CD44 is an established cancer stem cell (CSC) marker in several tumor types [7]; for example, the CD44 and CD271 in human head and neck squamous cancer cells [8] and the CD44 and CD133 in colorectal cancer cells [9]. These observations imply a central functional role in tumor biology when considering that CD44 is able to promote epithelial-to-mesenchymal transition (EMT) and therefore invasiveness and metastasis [10,11,12].

Although modifications of the extracellular domain of CD44 (such as O-/N-glycosylation, glycosaminoglycan substitution, and alternative splicing) regulate important functions of CD44, they cannot fully explain its critical involvement in such diverse cellular processes. Intracellular events also regulate CD44 actions, and this requires a functional intracellular domain. Given that CD44 lacks any intrinsic enzymatic (e.g., kinase) activity, it must be able to organize both membrane and cytosolic components and coordinate their functions. It is now evident that the short cytoplasmic tail of CD44 contains several structural motifs with the potential to selectively interact with cytoskeletal proteins and signaling effectors, often in cooperation with adjacent plasma membrane receptors. The assembly and coordination of such structural and signaling events can be regulated by post-translational modifications of the transmembrane and intracellular domains of CD44, such as palmitoylation, phosphorylation. and proteolytic processing.

## 2. Structural Features of CD44 ICD

The transmembrane and intracellular domains of CD44 are required for its proper membrane localization, ligand binding, and functions such as cell adhesion and migration [4]. For instance, mutation or deletion of the ICD results in the aberrant localization of CD44 within cellular membranes, an inability to bind HA, and impaired HA-mediated cell migration and tumor development [13,14,15,16]. Therefore, these highly conserved domains are essential for ligand binding and subsequent downstream intracellular events (i.e., outside-in signaling), probably by promoting the stabilization and clustering of CD44 receptors at the plasma membrane [4,13,17]. In contrast, their precise amino acid sequence is not a prerequisite for CD44 functions, since the replacement of either domain with equivalent domains from different adhesion receptors does not impair binding to HA, cell adhesion to HA matrices, or rolling interaction of lymphoid cells with HA pericellular coats [13,18], suggesting that the shape and conformation of these domains do not affect the inside-out signaling of CD44.

The 72-amino-acid-residue cytoplasmic tail contains two positively charged amino acid clusters in the juxtamembrane domain (^292^RRRCGQKKK^300^), which constitute the FERM-binding domain that mediates the interaction of CD44 with ERM (ezrin/radixin/moesin) cytoskeletal proteins [19,20]. This sequence also contains the putative acylation site Cys^295^, suggesting that partition of CD44 into lipid rafts may regulate CD44 association with ERM proteins (see Section 4, “Regulation of cytoskeletal organization and cell phenotype”). The FERM-binding domain is followed by the ankyrin-binding domain (^304^NSGNGAVEDRKPSGL^318^) [16], an additional cytoskeleton association site, the dihydrophobic basolateral targeting motif ^331^LV^332^ [21], and four C-terminal amino acids (^358^KIGV^361^) that represent the PDZ (PSD-95/Dlg/ZO-1)-domain-binding peptide [4] (Figure 1).

The CD44 cytoplasmic tail can be phosphorylated, and this phosphorylation is restricted to Ser residues [4,22]. Although it contains seven Ser residues (at positions 291, 305, 316, 323, 325, 336, and 337), only Ser^291^, Ser^316^, and Ser^325^ are subject to phosphorylation (Figure 1). Mutagenesis studies demonstrated that Ser^325^ is the primary site of constitutive CD44 phosphorylation, which is estimated to occur on approximately one-third of CD44 molecules and is mediated by Ca^2+^/calmodulin-dependent protein kinase II (CaMKII) [23]. The neighboring Ser^323^ residue, however, is not phosphorylated but is required for the binding of CaMKII to the receptor [24]. Functionally, mutations at Ser^325^ impair HA-mediated cell migration, but they do not affect the receptor’s HA-binding capacity [15,24,25]. The activity of CaMKII and subsequent phosphorylation of CD44 ICD at Ser^325^ are regulated by the intracellular levels of Ca^2+^. The influx of Ca^2+^ and/or its release from the intracellular stores result in the increased cytosolic Ca^2+^, which binds to calmodulin, leading to the activation of CaMKII. Interestingly, CD44 can trigger mobilization of Ca^2+^ either by antibody-mediated receptor ligation or HA binding, leading to elevated levels of intracellular Ca^2+^ [26,27,28]. The constitutive CD44 phosphorylation at Ser^325^ suggests that CaMKII is maintained in an active state either through reduced phosphatase activity or kinase autophosphorylation that reduces the dissociation rate of the kinase–calmodulin complex, resulting in calmodulin trapping and constitutive active kinase even at low Ca^2+^ levels [23,29], implying that the regulation of CD44 phosphorylation at Ser^325^ is complex. Dephosphorylation of Ser^325^ followed by the concurrent phosphorylation of Ser^291^ and Ser^316^ by activated protein kinase C (PKC) reveals a dynamic regulation of CD44 phosphorylation, although the overall levels of CD44 phosphorylation remain almost the same [30]. Ser^316^ is not a direct substrate of PKC but is found in a predicted protein kinase A (PKA) consensus site [4]. It can be phosphorylated upon cell stimulation (for example, by chemotactic agents such as phorbol esters, which activate PKC), in contrast to resting cells that contain unphosphorylated Ser^316^ [22,31].

## 3. Hyaluronan Internalization and Interactions with the Cell-Trafficking Machinery

The dihydrophobic motif ^331^LV^332^ within the intracellular tail of CD44 targets the receptor to the basolateral membrane of polarized epithelial cells [21], similarly to other proteins such as the major histocompatibility complex (MHC) class II invariant chain, the lysosomal integral membrane protein (LIMP)-II, and others [32]. In contrast to these proteins, which undergo clathrin-dependent endocytosis mediated by the LV motif, CD44 has been shown to interact with the non-clathrin-dependent machinery whereby it internalizes its main ligand (HA), suggesting that the intracellular domain of CD44 allows for additional specific interactions that dictate the exclusion of CD44 from clathrin-coated pits. In chondrocytes, for example, CD44 mediates HA internalization through pathways independent from pinocytosis, caveolin, and clathrin [33,34,35].

The critical role of CD44 ICD for HA endocytosis and turnover is evident, since tail-less CD44 mutants have a shortened half-life, reduced stability at the cell surface, and weakened binding to HA and are unable to internalize HA [36]. There are several pathways suggested for CD44-mediated HA internalization. CD44–HA complexes either follow a pathway of trafficking to lysosomes for degradation or follow a pathway in which they are routed directly to the recycling endosomes for return to the plasma membrane. Ubiquitination of CD44 ICD by membrane-associated RING-CH (MARCH)-VIII ubiquitin ligase results in CD44–HA endocytosis and translocation of CD44 into EAA1-containing compartments and late endosomes/lysosomes for degradation [37], while HA is gradually degraded by the coordinated actions of β-endoglycosidase HYAL-1, β-exoglycosidases, β-glucuronidase, and β-N-acetylglucosaminidase [38]. Alternatively, CD44 can be recycled back to the plasma membrane by clathrin-independent sorting proteins such as Hook 1 and EHD proteins. Hook 1, a microtubule- and cargo-tethering protein, promotes microtubule-dependent sorting of CD44 away from the EAA1+ endosomes and into recycling tubules to end up in the plasma membrane [39]. In addition, our proteomic approach identified Eps15 homology domain protein 2 (EHD2), a lipid-raft-residing protein that couples endocytosis to the actin cytoskeleton, as a potential CD44-interacting protein that binds in vitro to the last 15 amino acids of the CD44 C-terminal tail containing the PDZ-binding motif ^358^KIGV^361^ [40], suggesting that CD44 harbors internalization signals in its cytoplasmic tail that are recognized by endocytic adaptors. This is in agreement with the observation that the acylation of CD44 at Cys^286^/Cys^295^ and its resulting localization into lipid rafts is crucial for HA uptake [35,36,41]. More recently, EHD1 was found to interact with CD44 ICD and promote the endocytic recycling of CD44, thus preventing its lysosomal degradation and increasing CD44 cell surface expression [42]. EHD1 acts as a director of the endocytic trafficking and recycling of internalized cargos [43]. The authors suggested that EHD1 is a key regulator of the Hippo signaling pathway and its downstream target genes (such as SP1) as well as CD44 endocytic recycling and stability. According to their findings, the EHD1/CD44/Hippo/SP1 axis and the concomitant deactivation of Hippo enhance the CSC-like properties, EMT, and metastatic potential of lung adenocarcinoma cells [42].

Interestingly, the interactions of CD44 with cargo transport proteins also contribute to sustained receptor tyrosine kinase (RTK) signaling. One such protein is Rab7A, which switches between active Rab7A-GTP and inactive Rab7A-GDP forms. In its active GTP-bound form, Rab7A plays an essential role in epidermal growth factor receptor (EGFR) degradation by regulating its endocytic trafficking to the late endosomes/lysosomes. Wang and colleagues demonstrated that CD44 preferentially binds to the active Rab7A-GTP form and accelerates the formation of inactive Rab7A-GDP, thus inactivating Rab7A activity and inhibiting EGFR degradation [44]. These observations raise the possibility of a more general regulatory role of CD44 on RTK signaling, since CD44 is a negative regulator of Rab7A, which promotes endocytosis-mediated degradation of multiple RTKs.

## 4. Regulation of Cytoskeletal Organization and Cell Phenotype

In migrating cells, CD44 shows a preferential distribution in actin polymerization regions such as the lamellipodia, filopodia, and apical microvilli, suggesting a role of CD44 in the regulation of actin cytoskeleton reorganization. This regulation is not direct, since CD44 ICD does not contain any binding sites for actin filaments but instead structural motifs such as the FERM (4.1 protein, Ezrin, Radixin, Moesin)-binding domain and the ankyrin-binding domain, which allow CD44 to connect and interact with the cytoskeleton. Specific clusters of basic residues (^292^RRRCGQKKK^300^) in the cytoplasmic tail of CD44 constitute the FERM-binding domain that interacts with ERM proteins and merlin/NF2 followed by a structural motif containing Ser^316^ (^304^NSGNGAVEDRKPSGL^318^) that binds the ankyrin cytoskeletal protein [4] (Figure 1).

The ERM proteins and merlin are binding partners for a number of transmembrane receptors (like ICAM, syndecans, L-selectin, and integrins) acting as cross-linkers between the plasma membrane and the cortical actin filaments [45,46,47,48]. The neurofibromatosis type 2 (NF2) gene encodes the merlin protein, which has tumor-suppressing functions through regulating Hippo signaling, as well as receptor tyrosine kinases and downstream signal transduction pathways [49,50,51]. ERM proteins and merlin show a similar domain organization, with the highest homology in the conserved three-lobe N-terminal FERM domain (head). The ERM and merlin contain extended C-terminal domains, in which ERM has an interaction site for F-actin, whereas merlin lacks a direct actin-binding site. Due to this reason, ERM and merlin may have distinct regulatory mechanisms in the organization of the actin cytoskeleton. Both ERM and merlin can form head-to-tail intra- and/or intermolecular associations, and they can switch between active and inactive conformations through phosphorylation/dephosphorylation events.

Phosphorylation of ERM proteins promotes their transition from the closed “inactive” conformation (head-to-tail self-association) to the open “active” conformation (dissociation of the three-lobed structure from the C-terminal domain) and their recruitment to the plasma membrane, where they bind membrane phospholipids such as phosphatidylinositol 4,5-biphosphate (PIP_2_). These interactions stabilize ERM proteins close to membrane-associated molecules such as CD44, which adopts a more open conformation at its ICD upon Ser^325^ phosphorylation [52]. Activated ERM binds to the FERM-binding domain of CD44 ICD phosphorylated at Ser^325^ (a constitutive phosphorylation site operated by CaMKII), forming a dynamic CD44-cytoskeleton association. This association can be disrupted upon PKC activation, which triggers the complete dephosphorylation of Ser^325^ followed by the phosphorylation of other Ser residues such as Ser^291^ and Ser^316^, leading to dissociation of the ERM proteins from CD44 and its disengagement from the actin cytoskeleton [4,30,53]. In particular, the phosphorylation switch from Ser^325^ to Ser^291^ by PKC results in the dissociation of the CD44–ezrin complex, probably because of the proximity of Ser^291^ to the FERM-binding domain, which is required for CD44-dependent chemotaxis [31]. However, treatment of human epidermal keratinocytes with IL-1β reduced Ser^325^ phosphorylation and promoted CD44 homomeric complexes, cytoskeletal association of CD44 through binding to ezrin, and extended monocyte-adhesive HA coats, thus enhancing monocyte retention on keratinocytes [54]. These results indicate cell type- and context-specific requirements for the cytoskeletal association of CD44 through post-translational modifications of its cytoplasmic tail, since Ser^325^ is not directly adjacent to the FERM-binding domain. The binding or exclusion of ezrin depends on its phosphorylation, but it may also involve other cytoplasmic binding proteins of CD44. For example, archaemetzincin-2 (AMZ2), a metalloproteinase of yet unknown function, is the only known protein that was identified by our proteomic approach to interact with a CD44 peptide containing phosphorylated Ser^325^ [40]. Furthermore, the proximity of Ser^325^ to the binding domain of another cytoskeletal protein, ankyrin, or the partition of CD44 into membrane lipid rafts would also regulate the CD44–ezrin association [55,56]. In contrast to the ERM proteins, it is the dephosphorylated form of merlin that binds CD44 ICD. Phosphorylation of merlin at its C-terminal tail generates the open conformation of the protein, resulting in the inactivation of its tumor-suppressing activity [57]. Several Ser/Thr kinases (including PKA, Akt, and p21-activated kinase/PAK) are involved in merlin phosphorylation, resulting in either the open “inactive” conformation or its degradation through the ubiquitin–proteasome system [58,59].

Therefore, the phosphorylation status of CD44 at specific Ser residues within the ICD is crucial for the dynamic association of the receptor with the cytoskeletal network and CD44-mediated chemotaxis, motility, and cell phenotypic changes. These events are not random but instead fine-tuned by the coordinated action of kinases and phosphatases. These enzymes could bind directly or indirectly to CD44. For example, ezrin interacts with PKC, thereby acting as a scaffolding protein for other signaling molecules to regulate phosphorylation of CD44 and associated proteins/receptors [60]. On the other hand, the existence of a PDZ-binding motif in CD44 ICD raises the possibility of a direct interaction of CD44 with PDZ domain-containing phosphatases or indirectly through PDZ domain adaptor proteins, which in turn bind protein phosphatases, thereby regulating the phosphorylation level of CD44 itself as well as other adjacent signaling molecules/receptors. For example, binding of syntenin, a syndecan-binding PDZ protein [61], to the C-terminus of protein tyrosine phosphatase (PTP) eta could result in the recruitment of this transmembrane PTP into syndecan-containing complexes [62,63]. Interestingly, PTP eta is negatively regulated by PKC, so their possible simultaneous recruitment into a CD44-containing complex could result in localized inactivation of the PTP, thereby affecting Tyr phosphorylation of other CD44-associated kinases like PDGFRβ, Met, EGFR, and Src. To identify PDZ domain-containing proteins and other proteins that could interact with the last 15 amino acid residues of CD44 ICD, which includes the PDZ-binding peptide (^358^KIGV^361^), we performed a proteomic approach using a peptide-based pull-down assay [40]. The analysis revealed the presence of the IQ-motif containing GTPase activating protein (IQGAP1). IQGAP1 interacts with cytoskeletal proteins, including F-actin, microtubules, and the small Rho GTPases Rac1/Cdc42/RhoA. IQGAP1 acts downstream of Rac1/Cdc42 and modulates the dynamic equilibrium in E-cadherin–β-catenin–α-catenin complexes at cell–cell junctions, thus modulating cell adhesion and migration [64,65,66]. IQGAP1 has been shown to inhibit the intrinsic GTPase activities of Rac1 and RhoA, resulting in the stabilization of their active GTP-bound forms, and it promotes the formation of RhoA-dependent F-actin stress fibers, suggesting an important role for IQGAP1 in the CD44-mediated reorganization of the actin cytoskeleton, migration, and proliferation in the presence of HA [67,68]. Other mechanisms involving the Rho-family GTPases also mediate CD44-induced cytoskeletal rearrangements. For example, association of the receptor with RhoA triggers the binding of ankyrin to the specific ankyrin-binding motif of CD44 ICD, which is required for anchorage-independent growth and HA-mediated cell migration [69].

The interaction of CD44 with the actin cytoskeleton through binding to ERM proteins and ankyrin is further stabilized by the partitioning of CD44 into detergent-insoluble, cholesterol-rich nano-domains (lipid rafts) of the plasma membrane [4,69,70] (Figure 1). The highly conserved intramembrane domain of CD44 can be reversibly palmitoylated at Cys^286^ and Cys^295^, which enhances localization of the receptor in cellular membranes as well as its interactions with cell surface and cytoplasmic proteins, such as several RTKs, innate receptors (TLRs), ABC transporters, EMMPRIN, PI3K, Src kinases, ezrin, and hyaluronidase 2, which preferentially localize into lipid rafts [71,72,73,74]. Interestingly, the amount of CD44 in lipid rafts can vary in different cell types and can be displaced from these membrane domains upon E-cadherin expression, which negatively regulates HA–CD44 interactions and CD44-dependent tumor invasion and branching morphogenesis [75]. In immune cells, CD44 is directed into lipid rafts via palmitoylation, which impairs the CD3-mediated signaling [76]. The strong association between CD44 and the actin filament network causes CD44 to act like a transmembrane picket that connects the cytoskeleton to pericellular milieu, forming barriers against diffusion of molecules through the plasma membrane [77]. Furthermore, the specific location of ERM proteins determines their functions. For example, moesin can be released outside the cells, and intracellular/extracellular moesins can have opposing effects on pro-tumorigenic signaling [78]. In particular, in contrast to intracellular moesin, extracellular moesin exerts a tumor-suppressive effect by regulating cell adhesion through inhibition of Src and β-catenin signaling via its interaction with the CD44 extracellular domain and FN1.

Of particular interest is the cross-linking of CD44 receptors, which has been associated with the metastatic potential of several human cancers. Receptor aggregation can be drastically induced by both extracellular (pericellular HMW HA) and intracellular (phosphorylated moesin/ezrin) cues [79]. The association of CD44 with ezrin and actin promotes HA binding to cells [80,81]. Furthermore, it is likely that clustering of CD44 would enhance HA’s binding affinity compared to the weak affinity of single CD44 molecules to HA [82,83]. Therefore, CD44 homomerization as a result of its association with cytoskeletal ERMs (i.e., ezrin) could create tight connections with extracellular HA polymers. The resulting tight HA–CD44 complexes would allow free ends of HA to extend far from cell surface to expand the HA coat [54]. In addition, CD44 clustering on neutrophils and interactions with E-selectin on activated endothelial cells have been established as a prerequisite for neutrophil tethering and subsequent rolling arrest and emigration into perivascular tissues. This process seems to be enhanced by the interaction of CD44 ICD with ankyrin, since disruption of the ankyrin-binding site impaired HA–CD44-dependent rolling of neutrophils on E-selectin and CD44-mediated activation of Src kinases [84].

## 5. Regulation of Cell–Cell Contact Inhibition and Cell Growth

CD44-mediated contact inhibition of cell growth is modulated by counteracting CD44 ICD interacting proteins; e.g., the tumor suppressor protein merlin and ERM proteins. Their reversible dynamic association with CD44 ICD regulates the effects of the receptor on cell growth and motility since it provides an ON (CD44–ERM)/OFF (CD44–merlin) mechanism in the CD44–actin cytoskeleton association and signal transduction pathways (Figure 1). The importance of these interactions in cell physiology is evidenced by the finding that long-lived naked mole rats, which synthesize HA of exceptionally high molecular size, show a remarkable resistance to cancer through regulation of cell–cell contact inhibition via the HA–CD44–merlin signaling axis [85,86,87]. Mechanistically, ERM proteins and merlin are both hypophosphorylated in high-cell-density conditions [88]. Under these conditions, ERM proteins are inactive, while merlin adopts the active unphosphorylated closed conformation and interacts with HA-bound CD44 and E-cadherin, thus stabilizing the homophilic E-cadherin interactions and cell–cell adhesion complexes and rendering the cells stationary. Activated merlin prevents proteolytic processing of the CD44 extracellular domain, which in turn preserves cell density signaling and inhibits cell proliferation and migration [89]. Furthermore, dephosphorylated merlin suppresses Ras-mediated signaling [89]. Merlin also couples cell density with Hippo pathway components [90,91]. Involvement of the Hippo pathway in contact inhibition was evident from the observation that the overexpression of the YAP pro-oncogenic transcriptional coactivator, a Hippo signaling pathway component, antagonizes contact inhibition [92]. The interaction of merlin with the cytoplasmic domain of CD44 promotes the recruitment of LATS (a Hippo pathway core Ser/Thr kinase), thus stimulating Hippo signaling and contact inhibition [93]. Reduced cell density induces p21-activated kinase (PAK)-mediated phosphorylation of merlin, leading to its dissociation from CD44 ICD and release into the cytoplasm, thus abrogating the growth-suppressive actions of merlin [94,95]. On the other hand, phosphorylated active ERM proteins bound to F-actin are recruited to the ICD of CD44 isoforms containing v6 exon (for example, CD44v6), which act as co-receptors for the RTK Met, promoting hepatocyte growth factor (HGF)-mediated activation of Ras and downstream signaling via the Erk MAP-kinase and PI3K/Akt pathways [96,97]. These observations imply a functional coordination of the CD44 stem region’s structural features with CD44 ICD to induce specific signaling pathways that drive cell growth and survival.

In addition to merlin, PAR1b (partitioning defective 1b) also links CD44 and the Hippo pathway [98]. Ooki and colleagues found that HMW–HA-mediated CD44 clustering induces the interaction of CD44 ICD with the PAR1b/microtubule affinity-regulating kinase 2 (MARK2) complex, which normally inactivates MST1 and MST2 Ser/Thr kinases through binding and subsequent inhibitory phosphorylation, thus triggering Hippo signaling activation and contributing to contact inhibition of cell growth. In contrast, LMW HA competes with HMW HA for binding to CD44, thereby inhibiting HMW–HA–CD44-mediated PAR1b recruitment and subsequent Hippo signaling activation [98].

Our proteomic analysis revealed the in vitro interaction of the inhibitor of apoptosis-stimulating protein of p53 (iASPP) with the C-terminus of CD44 [40]. iASPP inhibits p53-mediated cell death by suppressing its transcriptional activity on the promoters of Bax and PIG3 [99,100]. iASPP belongs to the family of ASPP proteins and was originally described as an ReIA-associated inhibitor (RAI) that inhibits the transcriptional activity of the NF-kappaB subunit p65 [101,102]. We identified the ankyrin-binding domain of CD44 ICD as the primary site of CD44–iASPP interaction, which differed between normal and cancer cells [103]. The formation of iASPP–CD44 complexes was induced by HA stimulation, while iASPP was found to be important for HA-induced, CD44-dependent migration and adhesion of normal fibroblasts [103]. Importantly, in the absence of iASPP, CD44 interacted with p53 and triggered the production of intracellular reactive oxygen species (ROS), suggesting an important role of CD44 in p53-mediated apoptosis. Considering the fact that p53 binds to the CD44 promoter via a non-canonical p53 consensus sequence and suppresses CD44 expression promoting cell apoptosis [104], our findings suggest that balanced CD44–iASPP and iASPP–p53 complexes may regulate cell survival. The regulation of these events seems to be complex and cell-context-dependent, since the knock-down of CD44 led to a loss in contact inhibition, most likely through the Hippo pathway, and CD44 expression suppressed cell growth in fibroblast cultures [103].

## 6. Cleavage and Intracellular Release of CD44 ICD: A Master Regulator of the Transcriptome

The proteolytic cleavage of the extracellular domain or shedding of membrane-associated proteins is an irreversible post-translational modification that regulates cell–cell communication, intercellular signaling, and biological functions by releasing growth factors, enzymes, and soluble receptors [105]. In some instances, the remaining protein in the membrane is further cleaved, generating additional products that include a cytoplasmic fragment that can function in intracellular signaling, revealing an additional level of regulation of cellular functions. In line with this, CD44 can undergo successive proteolytic processing by a number of proteases, resulting in the generation of both extra- and intracellular bioactive fragments. These proteases are recruited and act at the cleavage site under specific prerequisites. The formation of CD44 dimers has been suggested to be critical [106]. Others include the phosphorylation status of specific Ser residues within ICD, since dephosphorylation of CD44 at the membrane proximal Ser^291^ was found to be essential for its ectodomain cleavage mediated by PKC [107], while selective binding of different cytoplasmic partners to ICD also affects this process. For example, binding of CD44 ICD to merlin reduces, whereas its interaction with ERM proteins (such as radixin) triggers CD44 shedding [106,108,109]. CD44 cleavage is induced by stimulation with several growth factors and cytokines as well with fragmented HA to activate a plethora of signaling pathways, including Ras, PKC, Rac1, and extracellular Ca^2+^ influx [110].

The sequential cleavage of CD44 begins with the ectodomain shedding by transmembrane MMPs (i.e., MT1-MMP and MT3-MMP) as well as other CD44 sheddases (i.e., disintegrin or ADAM 10, ADAM 17, and meprin β), resulting in the generation of a membrane-associated fragment of CD44 while its extracellular N-terminal region is released to the ECM [5,111]. ERM proteins are involved in CD44 shedding, since they mediate the colocalization of MT1-MMP and CD44. In particular, the radixin FERM domain simultaneously binds the cytoplasmic tails of MT1-MMP and CD44, forming a complex that is stabilized through anchoring to filamentous actin. This ternary complex facilitates the recognition of the CD44 stem region by the PEX domain of the MT1-MMP ectodomain, resulting in CD44 shedding [108,109] (Figure 2). More recently, the involvement of ubiquitin ligase tumor necrosis factor receptor-associated factor (TRAF) 4/6 in CD44 cleavage was demonstrated [112]. TRAF4/6 induces small GTPase Rac1-mediated activation of MMPs, enhancing CD44 ectodomain shedding and thus promoting the pro-tumorigenic effects of CD44 (Figure 2). It has been suggested that this is a cell-type-dependent process, since the relative expression levels of TRAF4 and TRAF6 in different cell types determine which of these two structural related TRAF isoforms mediates CD44 cleavage [112].

Shedding of the CD44 ectodomain triggers the subsequent intramembranous cleavage of the membrane-associated fragment via regulated intramembranous proteolysis (RIP) catalyzed by presenilin-1-dependent gamma-secretase activity, resulting in the release of a CD44β-like peptide in the ECM and the generation of a CD44 ICD fragment within the cytoplasm [113,114]. CD44 ICD can then function as an intracellular cargo of proteins already bound to the receptor, interact with additional cytoplasmic proteins as a result of its altered location, or translocate to the nucleus, where it functions as a master regulator of the transcriptome (Figure 2). The released ICD fragment contains a nuclear localization sequence (NLS) that enables ICD to be imported to the nucleus through the nuclear pore complex. The NLS is identical to that of the FERM-binding domain of CD44 ICD, which consists of two clusters of basic amino acids (^292^RRRCGQKKK^300^) [115] (Figure 1). Upon its translocation to the nucleus, the ICD acts as a transcription factor through binding to consensus sequences such as the 12-O-tetradecanoylphorbol 13-acetate (TPA)-response element or CCTGCG found in the promoter regions of *MMP-9*, *HIF-1 αlpha* genes as well as genes involved in cell metabolic pathways (see below) [116]. Of particular interest is the ability of CD44 ICD to induce the expression of full-length CD44 via a positive feedback loop through binding to the CBP/p300 transcriptional activator [110] (Figure 2). Other transcriptional activities of CD44 ICD that contribute to its oncogenic properties include its interaction with the CREB transcriptional factor, which induces CREB phosphorylation and binding to the promoter of cyclin D1, resulting in increased cyclin D1 expression and sustained cell cycle progression [117] (Figure 2). Moreover, CD44 ICD endows tumor cells with stemness and chemo/radio-resistance properties. Johansson and colleagues showed that CD44 ICD is released at hypoxia, interacts with HIF-2 alpha, enhances HIF target gene expression, and is required for hypoxia-induced stemness of perinecrotic and perivascular glioma cells [118]. Bourguignon and colleagues have demonstrated the interaction of CD44 ICD with Oct4, Sox2, and Nanog stem cell transcriptional factors that support self-renewal, clonal formation, and therapeutic resistance [119]. The nuclear translocation of the ICD–Oct4–Sox2–Nanog complex promotes their coordinated recruitment and binding to response elements in the promoter of target genes, thus enhancing their transcriptional activity and the acquisition of stem cell properties [120] (Figure 2).

## 7. Regulation of Cell Metabolism

One of the proteins that were identified by our proteomic approach to interact with the C-terminus of CD44 was the M2 splice isoform of pyruvate kinase (PKM2), a glycolytic enzyme that catalyzes the generation of pyruvate and ATP from phosphoenolpyruvate and ADP [40]. While this finding was considered to possibly be non-specific since PKM2 is an abundant protein that is often detected in biotin-based pull-down experiments, two years later, Tamada and colleagues confirmed the interaction of CD44 with PKM2 [121]. This finding suggested a role of CD44 in cancer cell metabolism, since PKM2 is a key enzyme that regulates aerobic glycolysis or the Warburg effect, a metabolic pathway utilized by tumor cells regardless of the local availability of molecular oxygen to produce energy (ATP) and promote tumor growth [122]. The low or high enzymatic activity of the PKM2 isoform dictates the conversion of pyruvate to lactate, which leads to the Warburg effect or to acetyl-CoA for mitochondrial oxidative phosphorylation, respectively [122,123,124,125]. Interestingly, the interaction of CD44 with PKM2 reduces the enzymatic activity of PKM2 via tyrosine phosphorylation by RTKs, which triggers glycolysis and the flux to the pentose phosphate pathway, a major source of NADPH (Figure 2). This mainly occurs in hypoxic or p53-mutated glycolytic cancer cells, since p53 is known to positively regulating mitochondrial oxidative phosphorylation through cytochrome c oxidase 2 induction [121,126]. Knock-down of CD44 enhanced oxidative phosphorylation, which led to GLUT1 suppression and a subsequent reduction in glucose uptake and pentose phosphate pathway flux. These changes resulted in reduced NADPH production, down-regulation of reduced glutathione (GSH) synthesis, and increased accumulation of ROS [121]. Moreover, other studies have shown that oxidative stress induces the oxidation of Cys^358^ in PKM2, which reduces PKM2 activity, promoting glycolysis and pentose phosphate pathway flux that results in the production of NADPH, GSH, and subsequent ROS depletion [127,128,129]. In line with these observations, CD44 regulates the activity of lactate dehydrogenase (LDH) that catalyzes the bidirectional conversion of pyruvate to lactate by modulating the ratio of LDHA and LDHB isoenzymes (Figure 2). In particular, ablation of CD44 lowered the LDHA-to-LDHB ratio, promoting mitochondrial respiration, while overexpression of CD44 increased the LDHA-to-LDHB ratio, enhancing lactate glycolysis and the Warburg effect in tumor cells [130]. It has been suggested that CD44 regulates the LDHA-to-LDHB ratio through the AMPKα/mTOR and HIF-1α signaling pathways [121,130,131]. These findings establish a key role for CD44 ICD in dictating the metabolic shift between mitochondrial oxidative phosphorylation and lactate glycolysis in tumor cells. These events provide tumor cells and likely CSCs with survival advantages while enabling them to bypass the cytotoxic effects of ROS and acquire therapeutic resistance properties.

Further, CD44 ICD enhanced aerobic glycolysis through induction of 6-phosphofructo-2-kinase/fructose-2, 6-biphosphate 4 (PFKFB4), a key enzyme that catalyzes 6-phosphofructose (F6P) to fructose-2,6-biphosphate (F-2,6-BP) via bidirectional conversion during glycolysis [116,132]. Given that PFKFB4 regulates the levels of F-2,6-BP, which is a potent allosteric activator of phosphofructokinase 1 (PFK1), a critical regulatory enzyme of the glycolytic pathway [133], CD44 ICD may be considered as a powerful modulator of glycolysis. It is suggested that CD44 ICD regulates PFKFB4 transcription through interaction with CREB that binds to the PFKFB4 promoter region (Figure 2). The resultant PFKFB4 promotes cancer cell stemness properties via up-regulation of aerobic glycolysis in cancer cells [132].

Additional important CD44 ICD-induced genes encoding enzymes involved in aerobic glycolysis include *ALDOC* (which encodes aldolase c, fructose biphosphate), and *PDK1* (which encodes pyruvate dehydrogenase kinase-1) [116], highlighting CD44 as a gatekeeper of the Warburg effect (Figure 2). ALDOC is a key enzyme of glycolysis and gluconeogenesis that catalyzes the reversible conversion of fructose-1,6-biphosphate to glyceraldehyde-3-phosphate and dihydroxyacetone phosphate, thus playing a crucial role in both ATP and glucose biosynthesis. PDK1 inhibits pyruvate dehydrogenase, which converts pyruvate to acetyl-CoA, thus forcing glycolysis to convert pyruvate into lactate rather than going into the tricarboxylic acid (TCA) cycle. The excessive production of lactate results in the generation of an acidified microenvironment that enables cells to degrade and invade the ECM, while NAD^+^ is regenerated to allow continued glycolysis [134] (Figure 2).

Other studies have provided data that link the cytoskeletal association of CD44 with systematic metabolic homeostasis. Specific deletion of ROCK1, a downstream kinase of the CD44/ankyrin/Cdc42 signaling pathway, attenuated obesity-associated insulin resistance in adipose tissue, indicating a role in obesity-related metabolic syndromes [135]. Moreover, p21-activated kinase 1 (PAK1), an additional downstream effector of the CD44/ankyrin pathway, is required for proper mitochondrial function and maintenance of cellular redox balance to prevent beta cell apoptosis, while it is involved in beta cell mass expansion and survival in islets of mice and humans during diet [136,137]. Interestingly, CD44–ERM interactions may regulate insulin secretion, since ERM-induced cytoskeleton rearrangement leads to glucose-stimulated insulin granule trafficking and secretion [138].

## 8. Conclusions

The CD44 tail has many more functions that (in the interest of space) are not discussed in the present review but are described in other excellent articles [4,5,139,140,141,142,143]. For example, CD44 is able to coordinate signaling responses, since many intracellular signaling molecules interact with CD44 ICD, including Rho-family GTPases and members of the Src family of non-receptor tyrosine kinases, which in turn activate PI3 kinase/Akt signaling [4,140]. However, there is still debate about the nature of these interactions and whether these are direct or indirect and are promoted by their co-localization in specialized cellular membranes and organelles. Notably, there is evidence to suggest that CD44v6/Met-driven signals may be generated within endosomal compartments, and internalization mechanisms have been identified [143]. The direct role of CD44 ICD in the co-receptor actions of CD44 has been established; for example, the FERM-binding domain is required for the scatter factor/hepatocyte growth factor (SF/HGF) to activate RTK Met downstream signaling [96].

Moreover, CD44 ICD may have roles during viral and/or bacterial infections. For example, CD44–ezrin interactions are involved in the internalization process of *Coxiella burnetii*, the etiologic agent of Q fever, in non-phagocytic cells [144]. The aberrant expression of CD44 together with the increased amount of HA triggered by the dengue virus predicted an increased risk of developing warning signs and thus severe dengue virus infection, resulting in perturbed vascular integrity and increased vascular leakage [145]. However, further studies are required to determine the exact role of CD44 ICD in viral/bacterial infections in vivo.

In conclusion, the association of CD44 ICD with cytoskeletal effectors (ERM, merlin, IQGAP1, and ankyrin) drives cytoskeleton rearrangements and affects the distribution of organelles and transport of molecules. In addition, through specific interactions of its cytoplasmic tail, CD44 regulates the cell-trafficking machinery, the transcriptome, and major cell metabolic pathways, with substantial impact on cell functional properties like survival, proliferation, adhesion, differentiation, therapeutic resistance, stemness properties, and EMT (summarized in Table 1). This long tale of the short tail of CD44 is still expanding, and it seems that we are still at the tip of the iceberg.

## Figures and Tables

**Figure 1 cancers-15-05041-f001:**
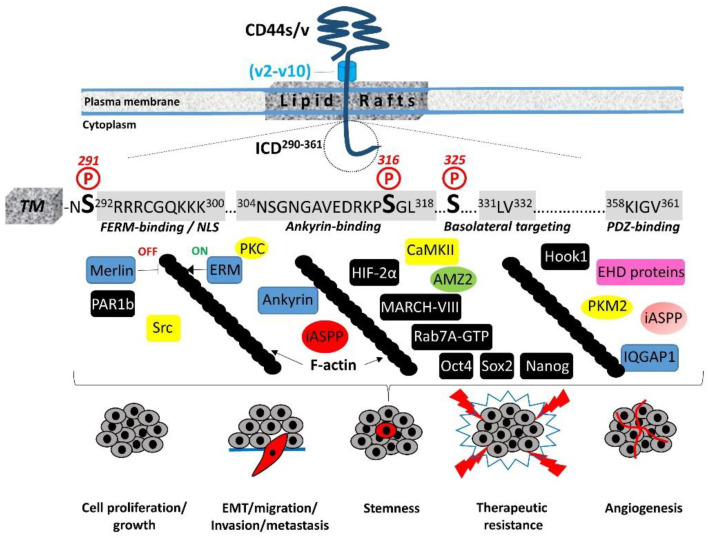
Structural motifs of CD44 ICD and their interactions with cytoplasmic proteins. Ser residues (Ser^291^, Ser^316^, and Ser^325^) that are subject to phosphorylation are highlighted. Cytoskeletal proteins (blue), kinases (yellow), and proteins involved in cell-trafficking machinery, metabolism, and transcription are shown (for details, see text). Proteins with yet uncharacterized interactions with specific domains/sites of CD44 ICD are indicated in black boxes. The functional importance of CD44 ICD interactions in (patho)physiological processes are also shown.

**Figure 2 cancers-15-05041-f002:**
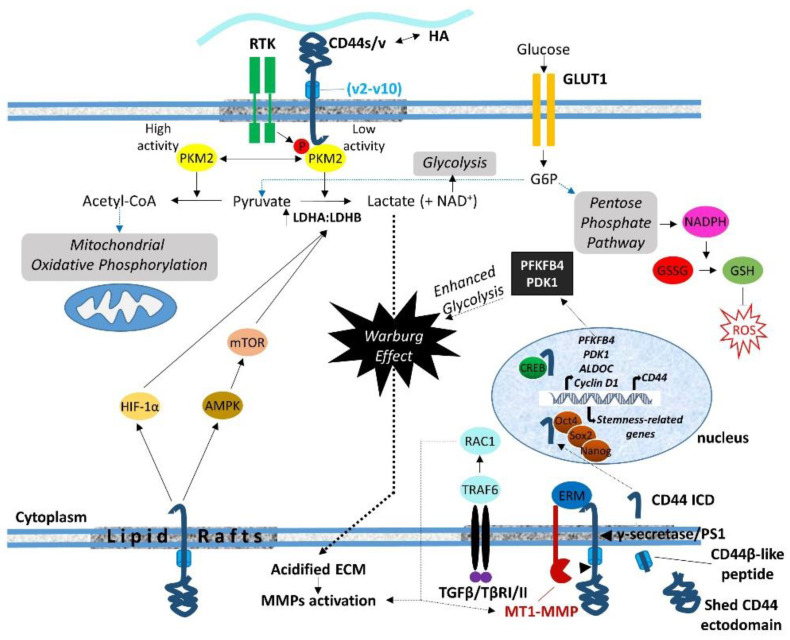
Schematic illustration of the roles of CD44 ICD in the regulation of transcriptome and cell metabolism. CD44 (via its ICD) dictates the metabolic shift from mitochondrial oxidative phosphorylation to aerobic glycolysis (the Warburg effect) through regulation of important metabolic enzymes such as PKM2, PFKFB4, and LDH. Enhanced pentose phosphate flux results in NADPH production that promotes GSH synthesis, which in turn suppresses ROS accumulation. Increased lactate production due to enhanced aerobic glycolysis results in the generation of an acidified pericellular microenvironment, which favors MMP activation. MMPs can be also activated by the TGFβ/TβR/TRAF6/RAC1 axis, thus triggering CD44 cleavage and the translocation of CD44 ICD to the nucleus, where it regulates the transcription of target genes, including CD44 itself (for details, see text).

**Table 1 cancers-15-05041-t001:** Importance of CD44 ICD in cellular pathways and functions.

Cellular Pathway	Cellular Function	CD44 ICD Interaction(s)	Ref(s)
Cell trafficking	HA endocytosis, pericellular matrix retention, and organization	Absence of ICD	[36]
	CD44–HA endocytosis and degradation	MARCH-VIII ubiquitin ligase	[37]
	CD44 endocytic recycling and stabilization, stemness, and EMT	Hook1, EHD1, and EHD2	[39,40,42]
	Sustained EGFR signaling and survival	Rab7A	[44]
Cytoskeletal organization	Growth, anchorage-independent growth, migration, adhesion, chemotaxis, and leukocyte binding	ERM, merlin, and ankyrin	[4,30,31,52,53,54,55,56,57,60,69,97]
	Migration and proliferation	IQGAP1	[40,67,68]
	Contact inhibition of growth, migration, apoptosis, and proliferation	Merlin	[85,86,87,89,90,91,93]
	Contact inhibition of growth	PAR1b	[98]
Transcription	Apoptosis, survival, migration, and adhesion	iASPP	[40,103]
	CD44 expression	CBP/p300 transcriptional activator	[110]
	Cell cycle progression	CREB	[117]
	Stemness and therapeutic resistance	HIF-2 alpha	[118]
	Stemness and therapeutic resistance	Oct4, Sox2, and Nanog	[119,120]
Metabolism	Survival, stemness, and therapeutic resistance	LDH, PFKFB4, ALDOC, and PDK1	[116,130,132]
	Survival and therapeutic resistance	PKM2	[40,121]

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
