# Peer review of "CD44 Intracellular Domain: A Long Tale of a Short Tail"

_cancers, 2023, doi:10.3390/cancers15205041_

Round 1

Reviewer 1 Report

The manuscript is generally good, the main problem is the basic English grammar, wordy sentences, lack of references for some statements. I have pointed out 45 comments based on the aforementioned problems, and can be seen in this attached PDF file. Note that: the small comment boxes need to be opened in order to see the comments.

See above attached.

Author Response

Thank you very much for the constructive criticism of this review article. I agree with almost all of the comments that have been pointed out. Please find the detailed responses and the corresponding corrections in the comment boxes found alongside the highlighted texts in the attachment.

Reviewer 2 Report

The author presented a comprehensive and profound study on modern knowledge of CD44 intracellular domain.

Comments

1.      Lines 174-176: The advantage of endocytic recycling versus lysosomal degradation of CD44 is not clear. This should be clarified.

2.      Lines 301-304; 501-504: These sentences are not clear. This should be corrected.

3.      Lines 392-397: These sentences repeat the text from paragraph 1 of the previous section. This should be corrected.

4.      Lines 445-449: How CD44 ICD induced stemness is associated is associated with ON (ERM) and OFF (merlin) regulation of CD44?

Author Response

Thank you very much for taking the time to review this manuscript. Please find the detailed responses below and the corresponding revisions highlighted in the attachment.

Point-by-point responses to Comments and Suggestions for Authors:

Comments 1: Lines 174-176: The advantage of endocytic recycling versus lysosomal degradation of CD44 is not clear. This should be clarified.

Response 1: By promoting the endocytic recycling of CD44, EHD1 increases CD44 cell surface expression and recycling. CD44 upregulation potentiates stemness and metastatic potential of tumor cells.

Therefore, I have accordingly revised the text (page 4, lines 174-176) to emphasize this point (Please see the attachment):

"More recently, EHD1 was found to interact with CD44 ICD and promote the endocytic recycling of CD44 thus preventing its lysosomal degradation and increasing CD44 cell surface expression [40]." 

Comments 2: Lines 301-304; 501-504: These sentences are not clear. This should be corrected.

Response 2: These sentences were accordingly modified (Please see the attachment):

301-304 (page 7): "In particular, in contrast to the intracellular moesin, extracellular moesin exerts a tumor-suppressive effect by regulating cell adhesion through inhibition of Src and β-catenin signaling via its interaction with CD44 extracellular domain and FN1."

501-504 (page 11): "Additional important CD44 ICD-induced genes encoding enzymes enzymes-encoding genes involved in oxidative glycolysis induced by CD44 ICD include ALDOC, which encodes aldolase c, fructose biphosphate and PDK1, which encodes pyruvate dehydrogenase kinase-1 [113], highlighting CD44 as a gatekeeper of the Warburg effect (Figure 2)."

Comments 3: Lines 392-397: These sentences repeat the text from paragraph 1 of the previous section. This should be corrected.

Response 3: Lines 392-397 (pages 8-9): This text has been accordingly corrected (Please see the attachment):

"For example, binding of CD44 ICD to merlin reduces, whereas its interaction with ERM proteins (such as radixin) (see below) triggers CD44 shedding [104, 106, 107]. CD44 cleavage is induced by stimulation with several growth factors and cytokines, as well as to stimulation by fragmented HA to activate a plethora of signaling pathways including Ras, PKC, Rac1 and extracellular Ca2+ influx [108]."

Comments 4: Lines 445-449: How CD44 ICD induced stemness is associated with ON (ERM) and OFF (merlin) regulation of CD44?

Response 4: Thank you for pointing this out. The ability of the ERM/merlin proteins to switch between an active and inactive conformation, together with the competition between the ERM proteins and merlin for CD44 binding, provides a mechanism to make and break the CD44-cytoskeletal association that could be involved in stemness properties and EMT.

As stated in text (lines 250-252, page 6), the phosphorylation status of CD44 at specific Ser residues within the ICD is crucial for the dynamic association of the receptor with ERM/merlin and, consequently, the cytoskeletal network and CD44-mediated chemotaxis, motility and cell phenotypic changes.

Reviewer 3 Report

Authors have presented a review emphasizing “CD44 Intracellular Domain: A Long Tale for a Short Tail”. Overall, the paper is well written however there are a few places where the necessary data is missing. However, these don't detract from the meaning of the text but a careful proofread could address these issues and improve the flow of the text. This manuscript could be a good addition to the literature related to CD44 importance and its various role. In my opinion, the manuscript can be published in this journal, after the authors have addressed the following minor issues:

·       Author is suggested to rewrite the abstract and elaborate the introduction part to align this with the theme of the manuscript.

·       Author must emphasize more in context to the main theme of the manuscript i.e. importance of CD44 intracellular domain and why author emphasize only this part. Is there any clinical study related to this.

·       Author is advised to improve the quality of all schemes.

·       Author must add one table depicting the importance of CD44 intracellular domain in various cellular pathways like cell-trafficking, transcription and metabolism that regulate cellular functions like growth, survival, differentiation, stemness and therapeutic resistance. current drug treatments of AD section and include a table that contains limitations.

·       Author is suggested to add section related to the role of CD44 intracellular domain in various cancer.

·       Check thoroughly the grammar and spell check properly.

Modify the conclusion part as per the suggestions and add future prospective section.

Minor English correction is required.

Author Response

Thank you very much for the constructive criticism of this review article. Please find the detailed responses below and the corresponding revisions highlighted in the attachment.

Point-by-point responses to Comments and Suggestions for Authors:

Comments 1: Author is suggested to rewrite the abstract and elaborate the introduction part to align this with the theme of the manuscript.

Response 1: I have accordingly revised the abstract (Please see the attachment). The introduction aims to briefly present the structural organization of CD44 and its extracellular ligands, while the last paragraph introduces the main theme of the manuscript that is the intracellular domain of CD44.

Comments 2: Author must emphasize more in context to the main theme of the manuscript i.e. importance of CD44 intracellular domain and why author emphasize only this part. Is there any clinical study related to this.

Response 2: CD44 has attracted a lot of attention the last years due to its critical involvement in almost every aspect of cell behavior. Moreover, the establishment of CD44 as a main CSC marker in various tumors implies a central functional role in tumor biology. This raises the important question of how and why a single receptor that is devoid of any enzymatic activity can mediate such diverse cellular processes. The present review aims to highlight the involvement of the less studied intracellular domain of CD44 in complex cellular responses that would potentially lead to the performance of clinical studies and new opportunities for therapeutic intervention. 

Comments 3: Author is advised to improve the quality of all schemes.

Response 3: Thank you for the suggestion. There has been a lot of effort to comply with the requirements of the journal. 

Comments 4: Author must add one table depicting the importance of CD44 intracellular domain in various cellular pathways like cell-trafficking, transcription and metabolism that regulate cellular functions like growth, survival, differentiation, stemness and therapeutic resistance. current drug treatments of AD section and include a table that contains limitations.

Response 4: I agree with this comment. Therefore, I have added one table (Please see the attachment, pages 21-22) depicting the importance of CD44 intracellular domain in various cellular pathways like cell-trafficking, transcription and metabolism that regulate cellular functions like growth, survival, differentiation, stemness and therapeutic resistance. 

Comments 5:  Author is suggested to add section related to the role of CD44 intracellular domain in various cancer.

Response 5: This review deals with the importance of CD44 intracellular domain in various cellular pathways like cell-trafficking, transcription and metabolism that regulate cellular functions like growth, survival, differentiation, stemness and therapeutic resistance. Therefore, the requested info regarding the role of CD44 ICD in various cancers is included in the existing sections. 

Comments 6: Check thoroughly the grammar and spell check properly.

Response 6: I agree with the comment. I have thoroughly checked grammar and syntax according to the comments of all reviewers. 
